


# Numerical investigation on the kinetic characteristics of the Yigong landslide in Tibet, China

Zili Dai[1], Fawu Wang[2], Hufeng Yang[3], Shiwei Qin[1]

[1]Department of Civil Engineering, Shanghai University, Shanghai, 200444, China
[2]Department of Geotechnical Engineering, Tongji University, Shanghai 200092, China
[3]Department of Geological Engineering, Southwest Jiaotong University, Chengdu 610031, China

*Correspondence to*: Zili Dai (zilidai@shu.edu.cn)

**Abstract.** To analyse the kinetic characteristics of a rapid landslide that occurred on 9 April 2000 in Tibet, China, a meshfree numerical method named Smoothed Particle Hydrodynamics (SPH) is introduced, and two-dimensional and three-
dimensional models are established in this work. Based on the numerical simulation, the landslide motion process is reproduced and the kinetic characteristics are analysed combined with the field investigation data. In the kinetic analysis, the landslide velocity, run-out distance, landslide accumulation, and energy features are discussed. Simulation results show that the landslide mass undergoes an acceleration stage after failure, then the kinetic energy dissipates gradually due to the friction and collision during the landslide propagation. Finally, the landslide mass blocks the Yigong river and forms a huge
landslide dam and an extensive barrier lake. The peak velocity is calculated to be about 100 m/s, and the run-out distance is approximately 8,500 m. The simulation results basically match the data measured in field, thus verifying the good performance of the presented SPH model. This approach can provide a new way to predict hazardous areas and estimate the hazard intensity of rapid landslides.

## 1 Introduction

Rapid landslides are a kind of catastrophic geological hazard which can cause very serious economic and human losses (Dai et al., 2019b). According to Huang (2007), about 80% large-scale rapid landslides in China occurred in the Tibetan Plateau. Among which, over 50 large-scale landslides were distributed along the Sichuan-Tibet Highway (Shang et al., 2003a). Therefore, rapid landslides pose a serious threat to the human engineering activities in Tibet, China.

Study on the kinetic characteristics of rapid landslides can contribute to the prediction of the impact area of the rapid
landslides and has recently attracted extensive attention of scholars around the world (Wei et al., 2019). Field survey combined with remote sensing technology is the most direct approach to obtain the basic dynamic characteristics and the impact area of rapid landslides. For example, Shang et al. (2003b) described the feathers of geological hazards along the Sichuan-Tibet Highway through field investigations. Fan et al. (2017) analysed the failure mechanism of the Diexi landslide occurred in Sichuan, China based on field investigations and satellite images. Karantanellis et al. (2020) proposed a





methodology based on the Unmanned Aerial Vehicle (UAV) to study two landslide cases within the territory of Greece. However, field investigation requires much manpower and financial resources, though it can directly obtain the first-hand information of the landslides.

A series of empirical-statistical models have been proposed to predict the run-out distances of landslides as a simplified analytical method. For example, Hunter and Fell (2003) presented a method to predict the post-failure behaviour for rapid

landslides in predominantly soil slopes. Huang and Cheng (2017) proposed a simplified model to predict the run-out of slope failures in landfills. Qiu et al. (2018) developed an empirical relationship to predict the travel distances of loess landslides in China. Su et al. (2019) presented an empirical model to predict landslide distance based on energy dissipation and mass point kinematics. The prediction results of the above models fit the measured data well. However, empirical model usually required some hypothetical conditions and it cannot consider the 3D complex terrain of a landslide and energy dissipation

during its propagation.

Physical model tests are an effective approach to investigate the failure mechanism and post-failure behaviour of landslides. For example, Zhou et al. (2020) carried out a large-scale physical model test to study the displacement characteristic of landslides. Pu et al. (2020) designed a shaking table test to investigate the dynamic response of loess slopes under the rainfall and earthquake coupling effect. Xie et al. (2020) performed series of model tests to investigate the sliding behaviour of

rotational landslides. Normally these model tests were in small-scale, and size effect is the main limitation. Tang et al. (2020) designed a large-scale centrifuge test model to investigate the deformation characteristics of reservoir landslides. However, these tests are very expensive and time-consuming.

With a rapid development of computer technology, a various of numerical methods have been developed and widely applied in all fields. Numerical simulation is becoming an important approach for landslide research, especially its stability analysis,

failure mechanisms and motion processes. For instance, mesh-based methods, such as the finite difference method (FDM) and the finite element method (FEM) have been widely applied in slope stability analysis (Ding et al., 2012; Chen et al., 2019), failure mechanism analysis (Maihemuti et al., 2016; Xiong et al., 2018), and failure process simulation (Bernander et al., 2016; Tang et al., 2017). Recently, mesh-free methods have been rapidly developed and widely applied due to their unique capability and advantages to handle the problems with large deformation and free surface. For example,

Discontinuous Deformation Analysis (DDA) and Discrete Element Method (DEM) have been widely applied to simulate the motion process of flow-like landslides (Liu et al., 2020; Peng et al., 2020; Zhu et al., 2020). Besides, Particle Flow Code (PFC) has been often used for the runout behaviour simulation of catastrophic landslides (Wei et al., 2019; Wang et al., 2020). Smoothed Particle Hydrodynamics (SPH) is a mesh-free method which has been proved to be applicable for kinetic characteristics analysis of flow-like landslides (Pastors et al., 2015; Cuomo et al., 2016; Yu et al., 2018; Zhang et al., 2020).

In this work, the SPH method is applied to simulate the motion process of the Yigong landslide in both 2D and 3D. The kinetic characteristics during the propagation are analysed, including the sliding path, movement velocity, runout distance,



accumulation, and energy features. The simulation results are compared with the survey data in field, which shows that the SPH model can accurately analyse the kinetic characteristics of rapid landslides.

## 2 Geological setting and landslide features

### 2.1 Geological setting


On 9 April 2000, the Yigong landslide occurred at Zhamunong gully in Bomi County, southeastern Tibet, China. The geographical coordinates of the landslide are N 30°12′ 11″, E 94°58′03″ (Xu et al., 2012). Along the banks of Yigong river, the mountains are very high and steep, which are covered with thick snow over 4,000 m and with dense vegetation below 3,500m. The valleys in this area are very deep under the erosions of glaciers and rivers.


The rock masses in the study area are main granitoid rocks, which have experienced strong weathering and have been partially metamorphosed into slate and granitic gneiss (Shang et al., 2003b; Zhou et al., 2016). The surface of the slope is composed of quaternary loose colluvial deposits. Thick glaciers and snow covered the slope rock, which could decrease the shear strength of the geomaterial after melting and increase the weight of the sliding mass. Due to the collision between the Eurasian Plate and Indian Plate, active faults are well-developed in the Tibetan Plateau. Jiali Fault and Yigong-Lulang Fault,


two of the major active strike-slip faults (Lee et al., 2003), meet at the mouth of Zhamunong gully, as shown in Figure 1. Earthquakes frequently occurred in the study area. For example, 14 moderate earthquakes (Ms = 4.0–5.9) were recorded around the Yigong lake from 1980 to 1996. Therefore, the tectonic activities in this area caused the rock structure fractured, loosened, and weakened, which provided favourable conditions for the Yigong landslide occurrence.

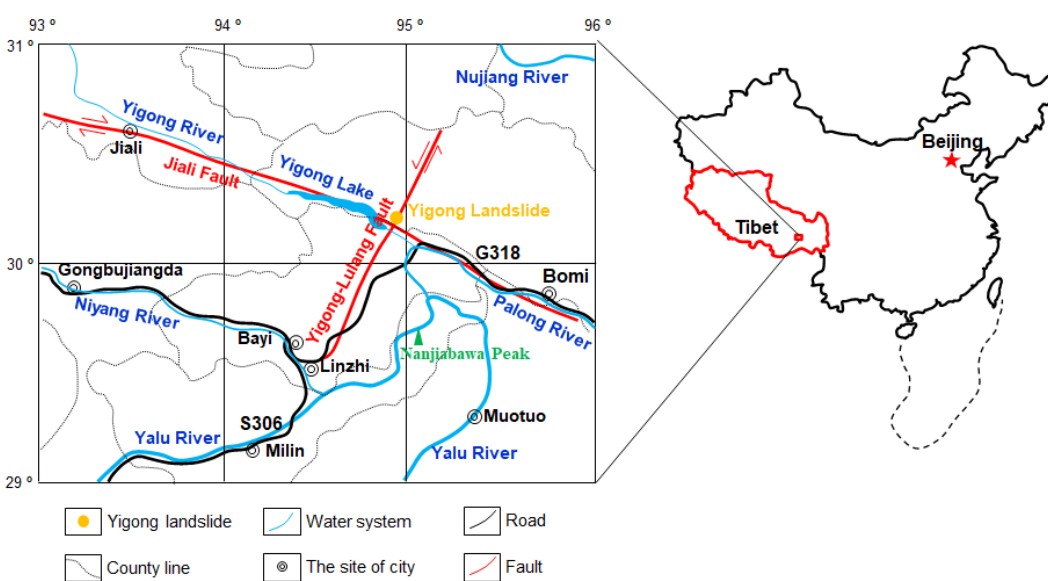

**Figure 1: Location of the Yigong landslide.**



The study area belongs to the temperate sub-humid plateau monsoon climate area. Influenced by the warm-wet air currents from the Indian Ocean, the weather is humid with clear four seasons and abundant in rainfall and sunshine. According to the local meteorological station, the annual rainfall averages 876.9 mm, and the cumulative sunshine hours is 1,544 hours. It was reported that the antecedent precipitation from 1 April to 9 April, 2000 was 42.9 mm, which was a main trigger of the landslide. Figure 2 is the average temperature in the study area during March and April in 2000. It shows that the average temperature gradually increased in that period, which might result in the glacier melting in the source area, thus may have increased the pore water pressure in the geomaterials and decreased the shear strength.

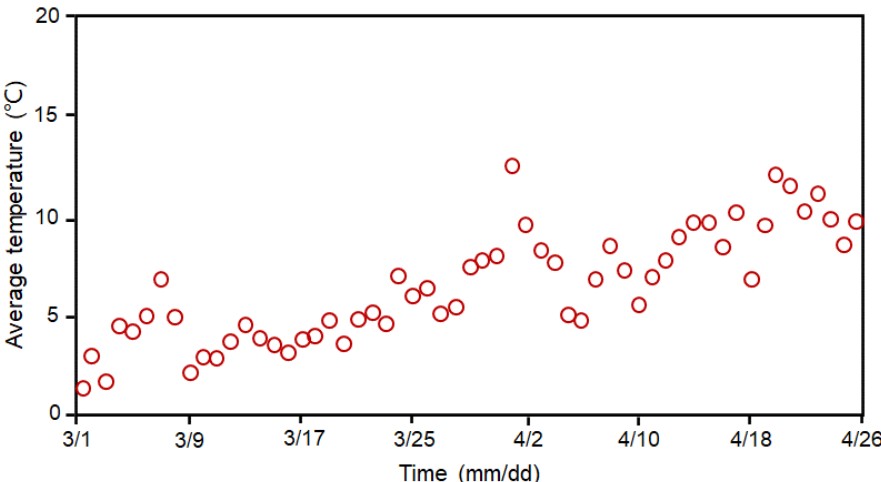

**Figure 2: Average temperature in the study area during 1 March to 30 April in 2000.**

**2.2 Landslide features**

Figure 3 shows an aerial view of the Yigong landslide occurred in Zhamunong gully (the base map is taken from Google Earth). About $3.0 \times 10^8$ m³ rock and soil slid down along the gully for about 3 min, the sliding direction is around 225°. The horizontal run-out distance is about 8,000 m, and the vertical dropdown is about 3,330 m from its source area at 5,520 m to its sediment fan at 2,190 m. The maximum sliding velocity of the landslide mass is higher than 100 m/s, and the average velocity is about 40 m/s.

Figure 4 is the topographic contour map of the Yigong landslide. The red dash line shows the landslide boundary, and the total surface area of this landslide was about 12.9 km2. The elevation of the landslide top is about 5,360 m, and the lowest elevation of the deposit area is about 2,200 m. Figure 5 shows the topographic map of the Yigong landslide in 3D. The slope at the both sides of the Zhamunong gully are very steep. Figure 6 shows the path profile of this landslide. The blue dashed line represents the estimated original slope surface before the landslide, and the green solid line is the present slope surface measured after the occurrence of the landslide. As shown in Figure 6, the landslide could be identified by three major zones: source zone, transfer zone and deposit zone. The characteristics of the three zones are described as below.



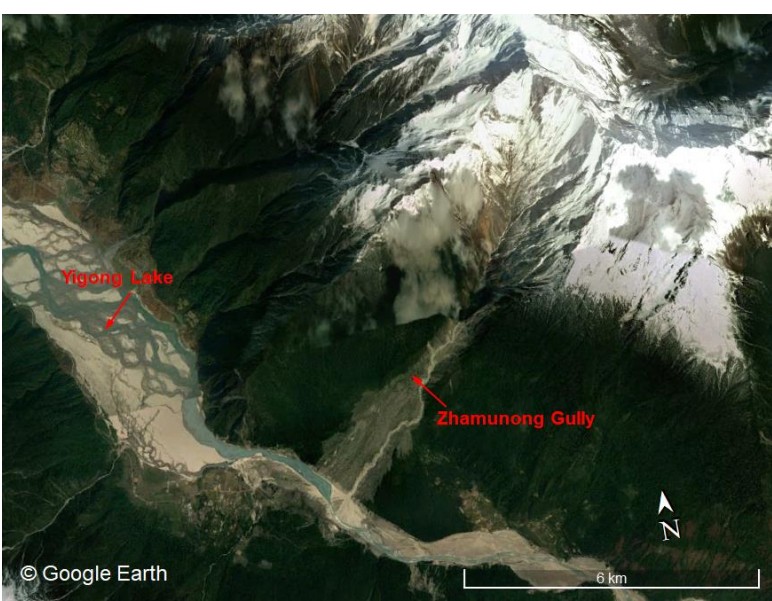

**Figure 3: Oblique view of the landslide © Google Earth.**


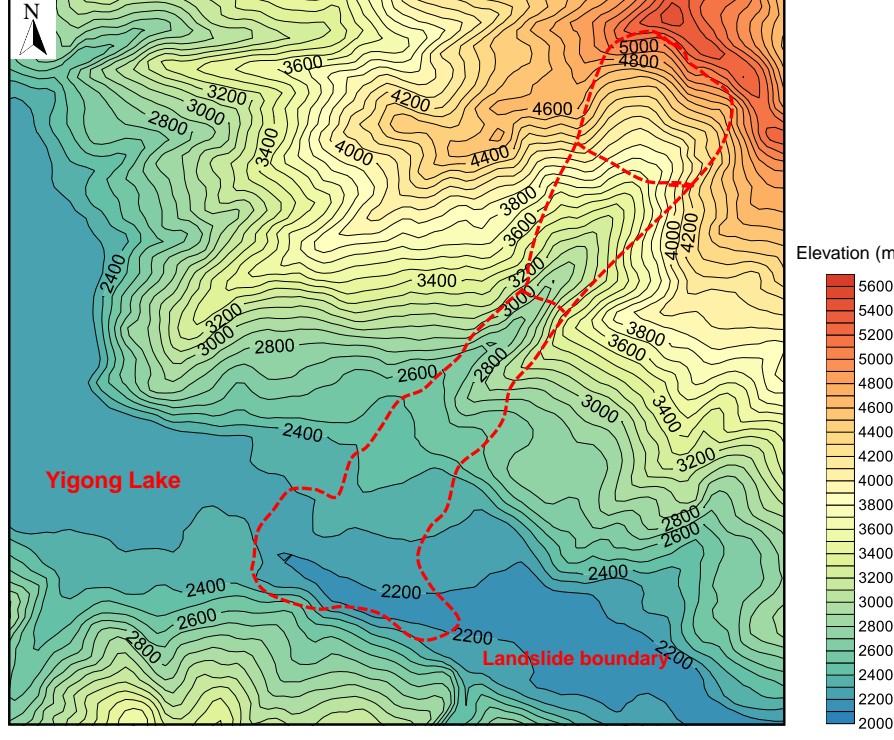

**Figure 4: Topographic contour map of the Yigong landslide.**




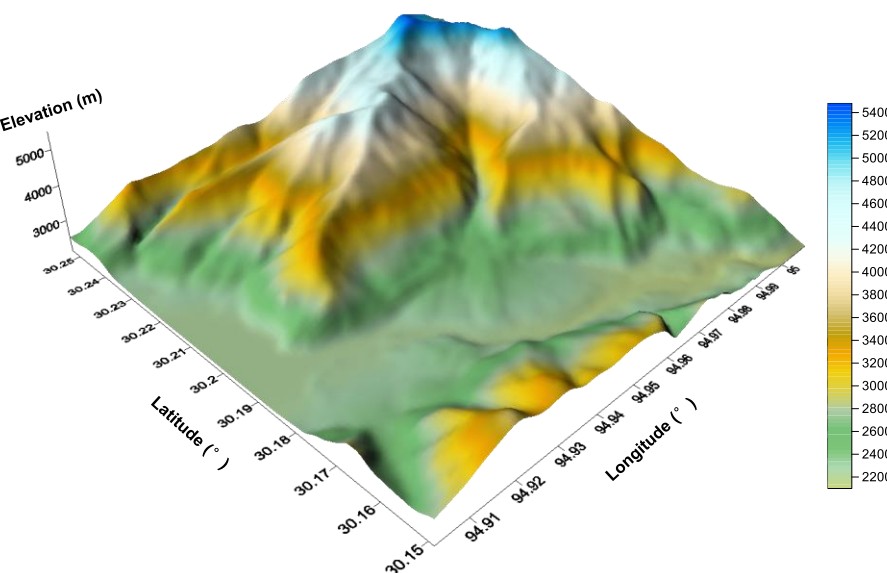

**Figure 5: Three-dimensional topographic map of the Yigong landslide.**


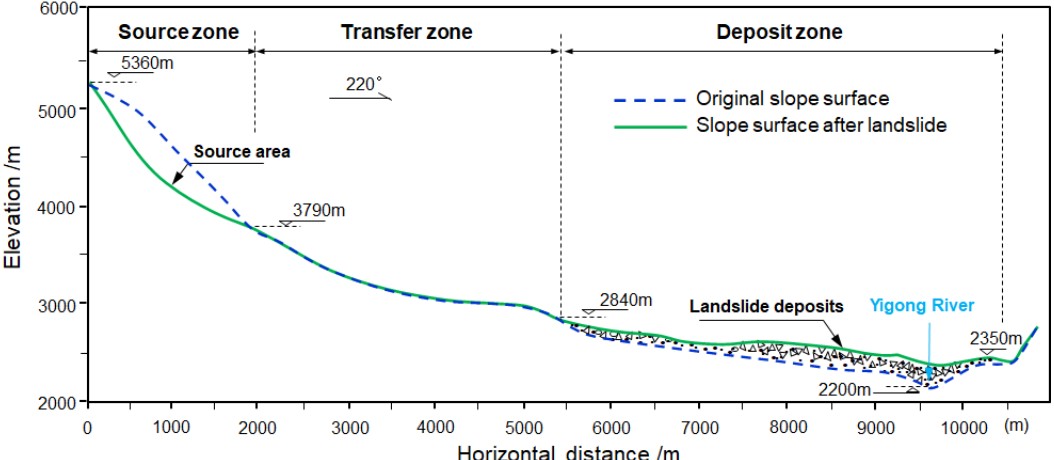

**Figure 6: Path profile of the Yigong landslide (Based on Yin, 2000).**

**2.2.1 Source zone**

Figure 7 shows an oblique view of the source zone of the Yigong landslide, which is located at the top of the Zhamunong
gully. It covered an area of about 1.40 km$^2$. The elevation of the source area sharply decreased from 5,360 m to 3,750 m,
with the slope angle of 40.0°. This area was covered by thick glaciers almost all the year round. The landslide mass was
wedge-shaped, wide at the top and narrow at the bottom. It poured down along the creek bed at a high speed.




### 2.2.2 Transfer zone

The transfer zone of this landslide covered an area of about 3.46 km². The axial length of this zone is about 3,200 m, and the
width ranges from 780 m to 1,500 m. The elevation of this zone ranges from 3,790 m to 2,840 m, with the height difference
of 950 m. The average slope of this zone is about 16.0°, which was much gentler than that of the source zone. As shown in
Figure 8, a lot of boulders were distributed in the gully. Most of them were angular with a diameter over 0.5 m.

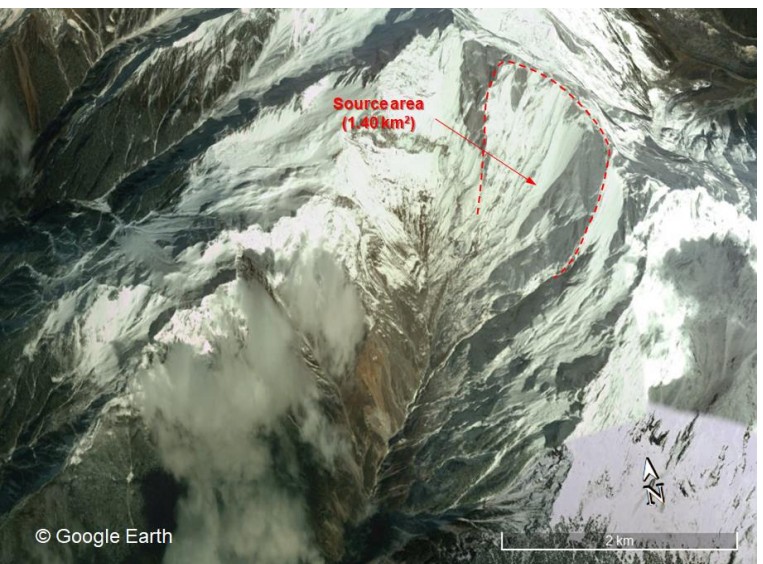

**Figure 7: Oblique view of the source area © Google Earth.**

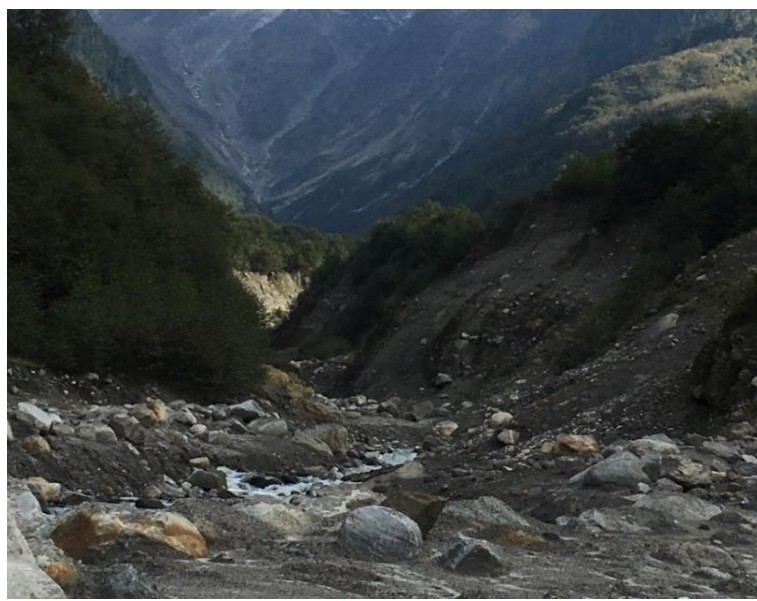


**Figure 8: Boulders in the transfer zone.**





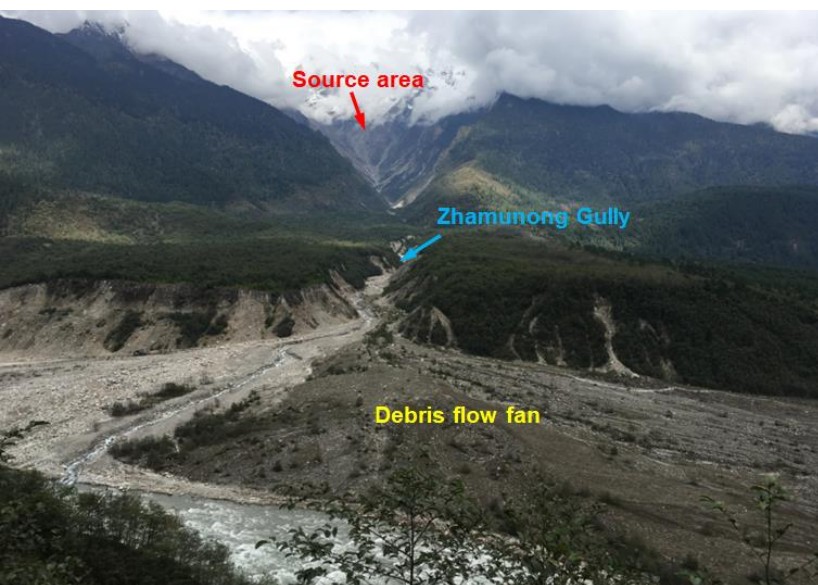

**Figure 9: Front view of the deposit zone (view direction NE).**

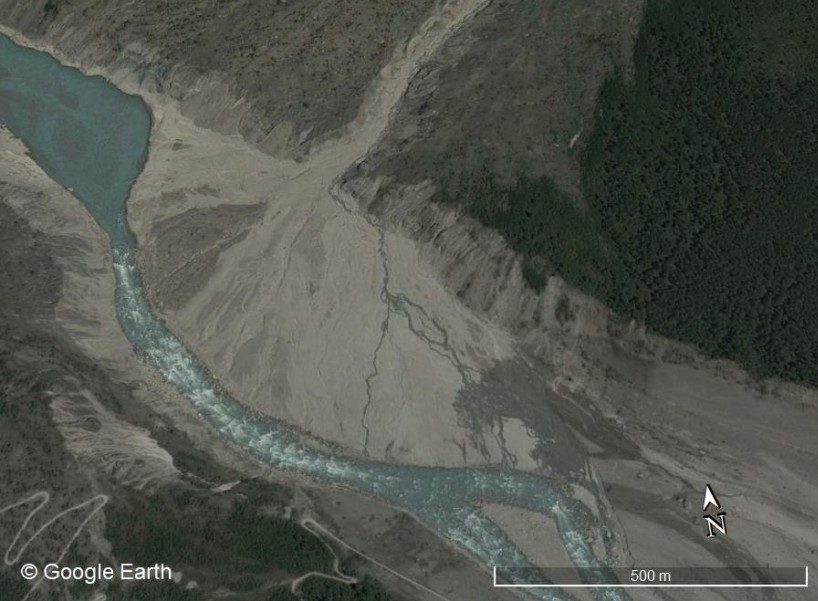

**Figure 10: Aerial view of the debris fan formed in the Yigong river channel © Google Earth.**

### 2.2.3 Deposit zone

Figure 9 is the front view of the deposit zone of the Yigong landslide. The elevation of this zone ranges from 2,200 m to 2,800 m, and the average slope is about 6.0°. The area of the landslide accumulation is over $6.0 \times 10^6$ m², and the average depth of sediment is about 50 m (Shang et al., 2003a). Due to the high motion velocity, the landslide mass flushed into the



Yigong river and formed a huge landslide dam and an extensive dammed lake. The length of the trumpet-shaped landslide

dam is about 4.6 km, the maximum width is 3.0 km, and the dam height is 60-120 m. The dam sloped at 5° at the upstream

side and 8° at the downstream side (Yin, 2000). After the landslide dam formation, water level of the Yigong lake

continuously rose at a rate of about 1 m/day, which flooded the Yigong tea farm, schools, and villages surrounding the

barrier lake. On 10 June 2000, the dam failed and resulted in a devastating flooding, which destroyed farms, villages, bridges,

and highways along its route. In recent years, the loose sediment was eroded by water from the Zhamunong gully and

formed a debris fan in the Yigong river channel, as shown in Figure 10.

## 3 Numerical model

To investigate the kinetic characteristics of the Yigong landslide, a meshfree numerical method named Smoothed Particle

Hydrodynamics (SPH) is applied, and two-dimensional and three-dimensional models are established for the rapid landslide

propagation simulation.

### 3.1 SPH algorithm

The SPH method was proposed in 1997 for astrophysical applications (Gingold and Monaghan, 1977). Recently, this method

has been widely applied to a large variety of engineering fields (Dai et al., 2017; 2019; Jamalabadi, 2020; Price and Laibe,

2020; Zhu and Zou, 2020). Compared to the mesh-based method, the major advantage of the SPH method is to bypass the

need for numerical meshes and avoid the mesh distortion issue and a great deal of computational work to renew the mesh

(Ma et al., 2020).

In the SPH method, the subject is represented by a set of particles to which the material properties such as velocity, density,

and pressure are associated. The properties are updated for each time step of the simulation following the conservation laws

of mass and momentum (Liu and Liu 2003).

In this study, the flow-like landslide is assumed as a kind of incompressible viscous fluid. Therefore, the continuity and

momentum equations are expressed by:

$$\frac{d\rho_i}{dt} = \sum_{j=1}^{N_i} m_j \left( v_j - v_i \right) \tilde{\nabla}_i W_{ij} \tag{1}$$

$$\frac{dv_i}{dt} = \sum_{j=1}^{N_i} m_j \left[ \frac{p_i}{\left(\rho_i\right)^2} + \frac{p_j}{\left(\rho_j\right)^2} + \delta \Pi_{ij} \right] \tilde{\nabla}_i W_{ij} + F_i \tag{2}$$

where $\rho$ is the particle density, $t$ is the time, and $m$ is the particle mass. $W$ is a smooth function, $v$ is the velocity and $F$ is the

body force. $\delta$ is the Kronecker delta and $\Pi$ is an artificial viscosity, which is used to improve the stability of the numerical

results (Monaghan and Gingold, 1983). $p$ is the pressure, which is calculated by an equation of state in this study:





$$p = \frac{\rho_0 c_s^2}{\gamma}\left[\left(\frac{\rho}{\rho_0}\right)^{\gamma} - 1\right] \tag{3}$$

where $\rho_0$ is the reference density, $c_s$ is the numerical speed of sound, and $\gamma$ is the exponent of the equation of state.

## 3.2 SPH model of the Yigong landslide

### 3.2.1 Material model


The landslide mass is a mixture of water, soil, and rock, which is complicated to describe. Hunger (1995) proposed a concept of "equivalent fluid", which is intended to simulate the bulk behavior of the moving mass. Pirulli (2010) used a frictional and a Voellmy rheology model to simulate the motion process of rapid flowlike landslides across three-dimensional terrain. Recently, some viscous fluid models have been widely used in the numerical modelling of flow-like landslides (Zhang and

Xiao, 2019). In the presented SPH model, the flow-like landslide mass is assumed as a Bingham fluid, and the relationship between the shear stress and the shear strain rate is defined as:

$$\tau = \left(\eta + \frac{p\tan\varphi + c}{\left(D_{II}\right)^{1/2}}\right)D \tag{4}$$

where $\tau$ is the shear stress of the fluid, $\eta$ is the yield viscosity coefficient in fluid dynamics, $p$ is the pressure which can be obtained by Eq. (3). $c$ and $\varphi$ are the cohesion angle and internal friction of the geomaterial. $D$ and $D_{II}$ are the strain rate and

its second invariant.

### 3.2.2 Boundary treatment

SPH method is ideal to deal with the free surface boundary. In the presented model, the free surface is identified through the criterion below:

$$\rho^* < k\rho_0 \tag{5}$$

where $\rho^*$ is the calculated density through the Eq. (1) and $\rho_0$ is the reference density. $k$ is the free surface parameter. When the particle is identified as a boundary particle, then zero pressure is applied.

For the solid wall boundary, ghost particles are placed on the boundary lines to exert repulsive forces and avoid the particles crossing the boundary. The velocities of the ghost particles are set to be zero to satisfy the non-slip boundary condition. Detail description of the non-slip boundary condition please refer to Dai et al. (2014).

### 3.2.3 Time integration

In a Lagrangian framework, the coordinates of each particles are updated at each time steps. A velocity-Verlet scheme is introduced in this SPH model to perform time integration.


$$X_{n+1} = X_n + V_n \Delta t + \frac{1}{2} a_n \Delta t^2 \tag{6}$$

$$V_{n+1/2} = V_n + \frac{1}{2} a_n \Delta t \tag{7}$$

$$V_{n+1} = V_{n+1/2} + \frac{1}{2} a_{n+1} \Delta t \tag{8}$$

where $X$, $V$ and $a$ are the displacement, velocity, and acceleration field, respectively.

## 4 Kinetic characteristics of the Yigong landslide

### 4.1 Two-dimensional modelling

According to the landslide profile in Figure 6, a two-dimensional SPH model is established in this study to simulate the post

failure behavior of the Yigong landslide, as shown in Figure 11. In this model, there are 13,568 particles in all, including 7,662 blue particles, which represent the landslide mass and 5,906 grey boundary particles, with a diameter of 8 m. The initial velocities of the particles are set to be zero. After slope failure, the landslide mass particles can slide down the slope under the action of gravity, while the boundary particles remain stationary throughout the simulation. The strength characteristics of the Yigong landslide mass were studied through a series of high-speed ring shear tests and rotary shear

tests in the previous researches (Hu et al., 2015; Wang et al., 2017). According to the test results, the values of the c and φ of the landslide mass can be approximately set as 10 kPa and 20°, respectively.

According to the simulation results, the motion process of the landslide mass takes about 200 s, which is basically consistent with the witnesses' description. Figure 12 presents the slope configurations at different points in time, which shows the motion process of the landslide mass after slope failure. The particles slide down from the top of the Zhamunong gully, and

then move along the steep slope by gravity. Finally, these particles run to an equilibrium state and accumulate in the Yigong river channel. The colour represents the velocity of the particles, which shows that the maximum sliding velocity of the landslide mass is about 100 m/s. To investigate the kinetic characteristics of the landslide, Figure 13 shows the time history curves of the landslide sliding velocity. The blue, red and green curves represent the velocity evolutions of the rear edge, front edge, and the average value of the landslide mass. The peak velocities are 102.6 m/s at the front edge, and 72.4 m/s at

the rear edge, respectively. The average velocity of the landslide mass during the landslide propagation is approximately 39.8 m/s. After slope failure, landslide body rapidly slides down and accelerates due to the gravity. In this stage, most of the potential energy of the landslide mass is converted into the kinetic energy. After the peak velocity, the kinetic energy is consumed due to the friction, collision, and the breakage of the sliding mass, and the velocity decreases gradually. The overall performance of the sliding mass is accelerated motion in the period 0–50 s and decelerated motion after 50 s. Figure

14 shows the comparison of measured and simulated landslide accumulation. The predicted landslide deposition zone is very



consistent with the measured data. The simulated runout distance is about 8,500 m, which can also match the measured result very well.

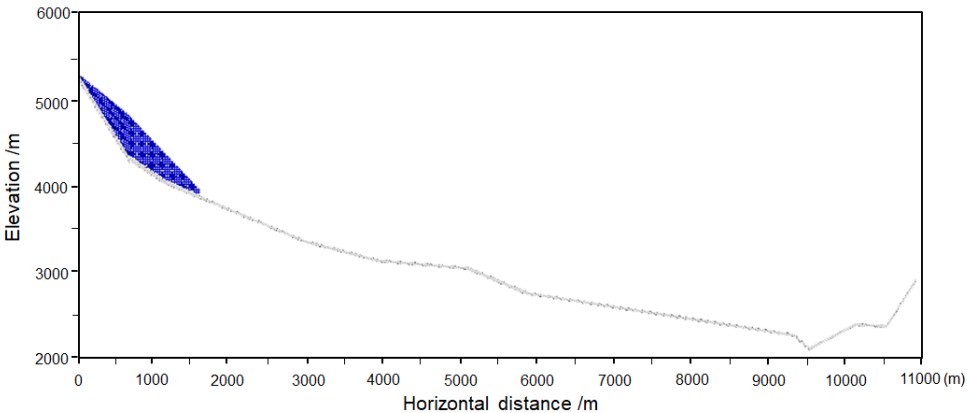

**Figure 11: Two-dimensional SPH model for the Yigong landslide.**


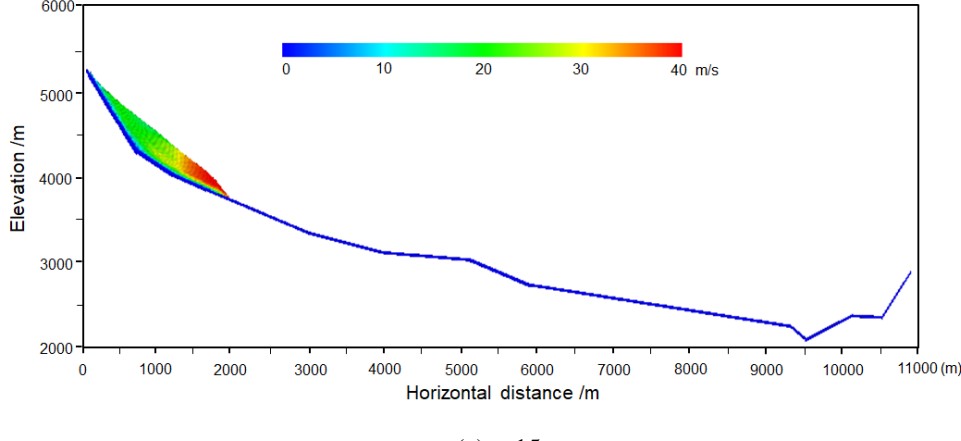

(a) t=15 s



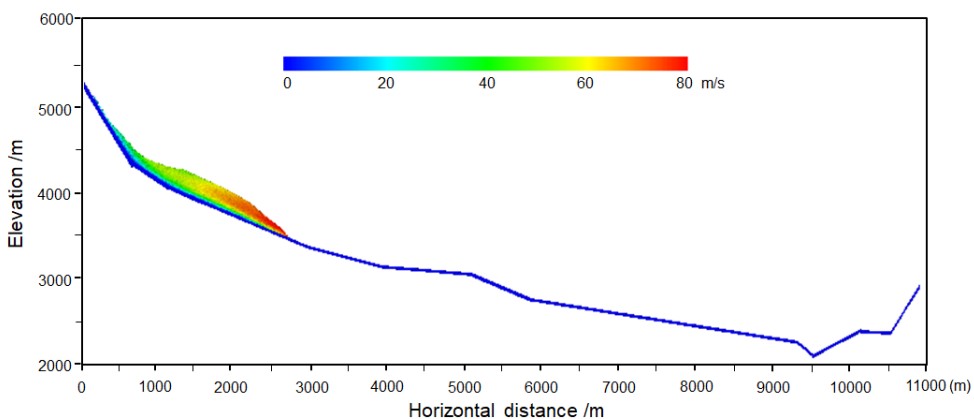


(b) t=30 s

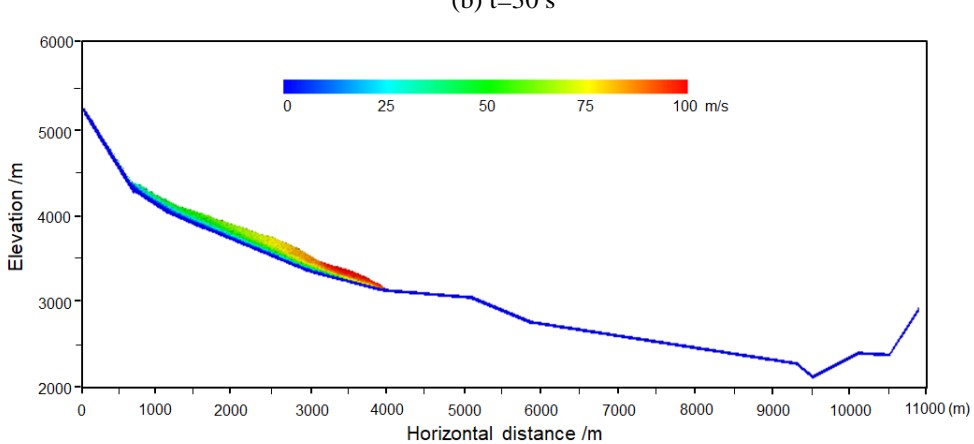

(c) t=45 s

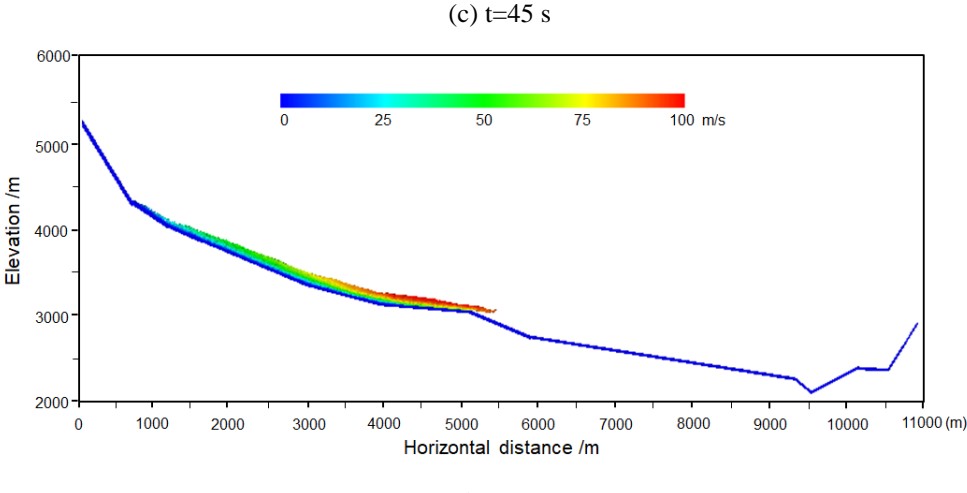

(d) t=60 s


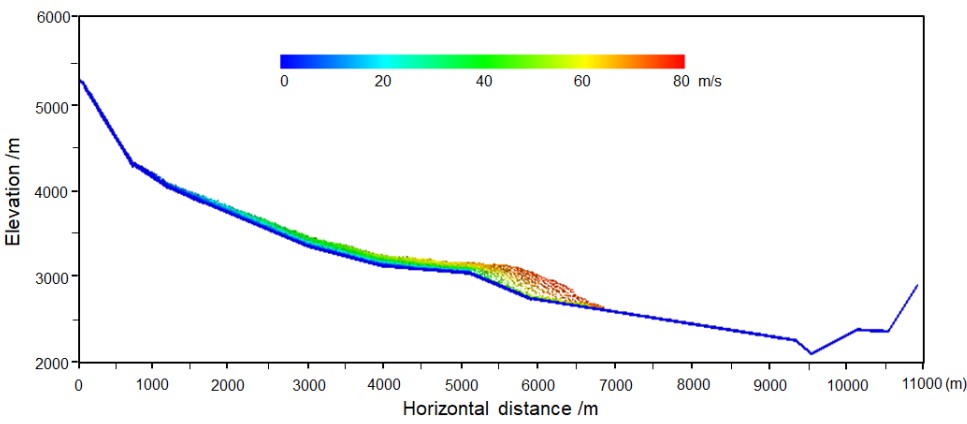


(e) t=80 s

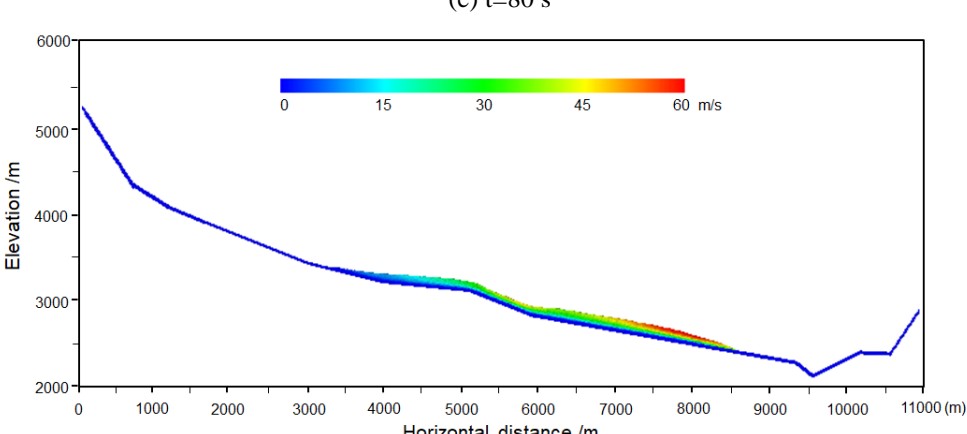

(f) t=120 s

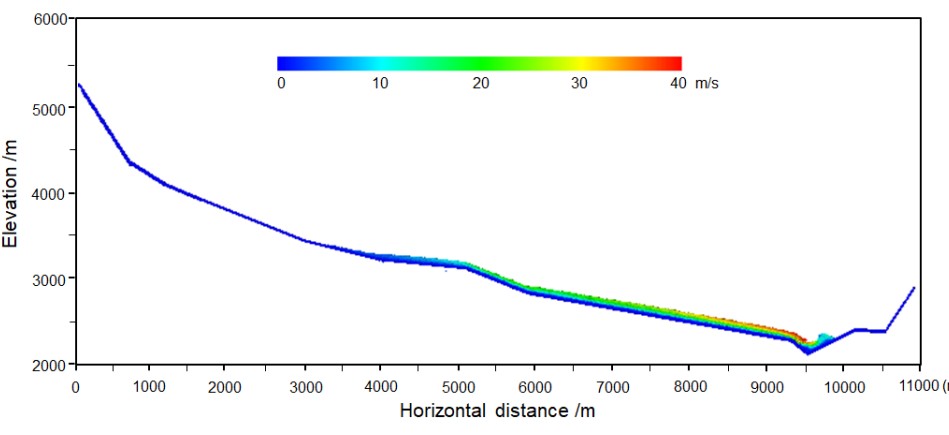


(g) t=150 s



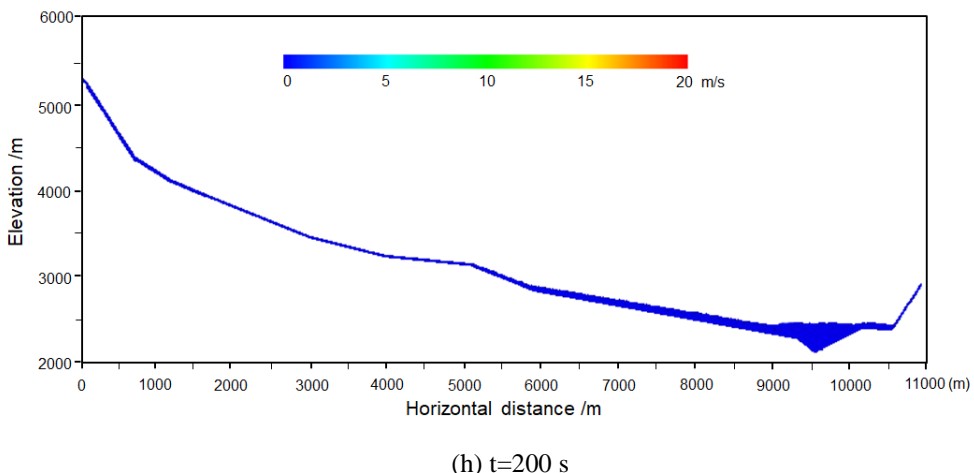

(h) t=200 s

**Figure 12: Two-dimensional simulated results for the motion process of the Yigong landslide.**

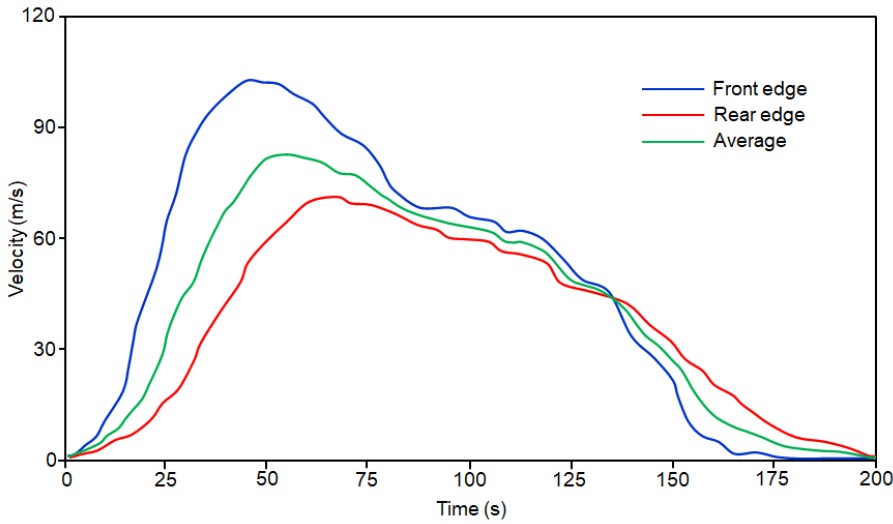


**Figure 13: Velocity time history curves of different parts of the landslide body.**




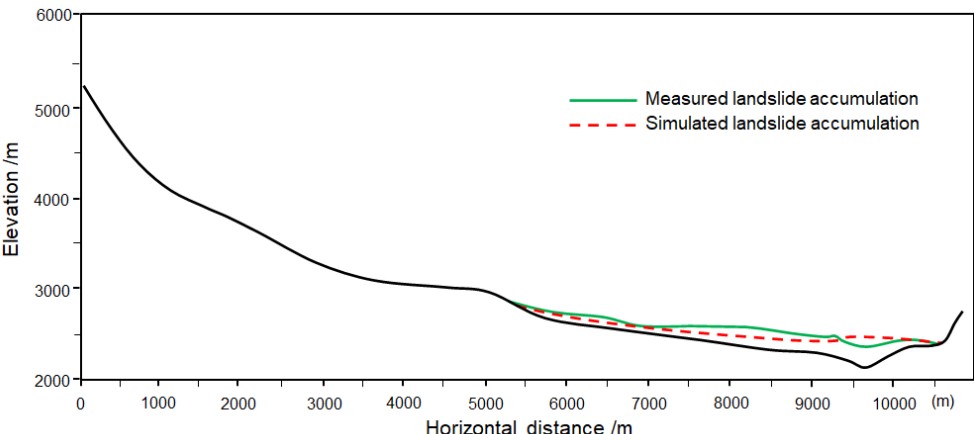

**Figure 14: Comparison of simulated and measured landslide accumulation.**

## 4.2 Three-dimensional modelling

To simulate the landslide propagation across 3D complex terrain, the 2D SPH model is developed into a 3D version. Based on the 3D topographic map shown in Figure 5, a 3D SPH model for the Yigong landslide is established. In this model, the landslide mass in the source area is discretized into about 11,000 particles with a diameter of 20 m. The number of particles along the vertical direction varies in different position according to the depth of the sliding surface in this position. The strength parameters used in 3D simulation are the same as those used in 2D model. Based on this model, the numerical

modeling of the Yigong landslide motion across 3D terrain is conducted and the results is shown in Figure 15. The color of the particles in the figures represents the sliding velocity. After slope failure, the landslide mass goes through an acceleration process since the slope is quite steep in the source area. The maximum sliding velocity is about 98.4 m/s which appears at 47.5 s after the slope failure. Afterwards, the landslide mass slows down gradually due to the friction and the collision during the propagation. Finally, the landslide mass crashes into a mountain on the opposite bank of Yigong river and then blocks the

river channel. The whole motion process takes about 200 s, and the final depositions of the landslide mass on the runout path are shown in Figure 15(g). The simulated maximum thickness of the landslide deposition is about 120 m, which is consistent with the field measurement result. Figure 16 shows the Yigong landslide accumulation. The shape of the simulated deposition zone is basically in agreement with the observed one.




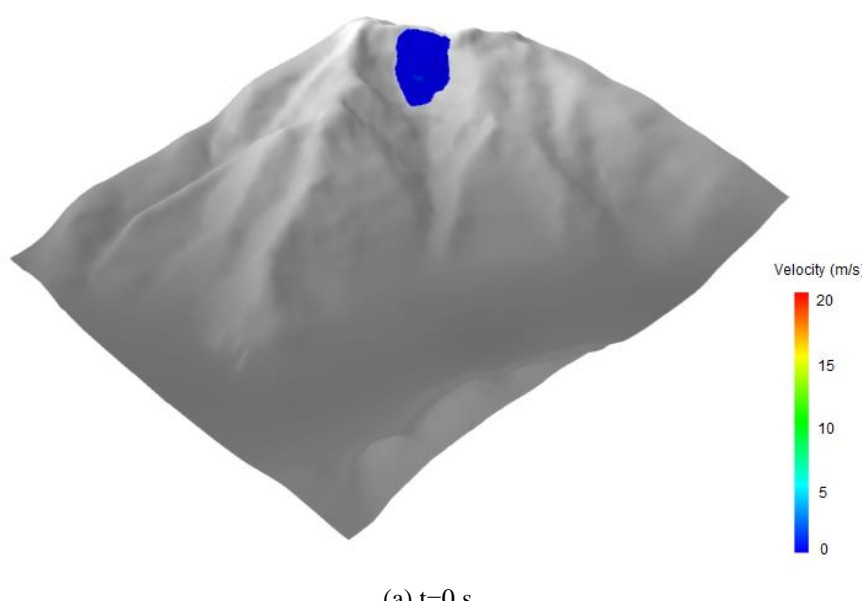


(a) t=0 s

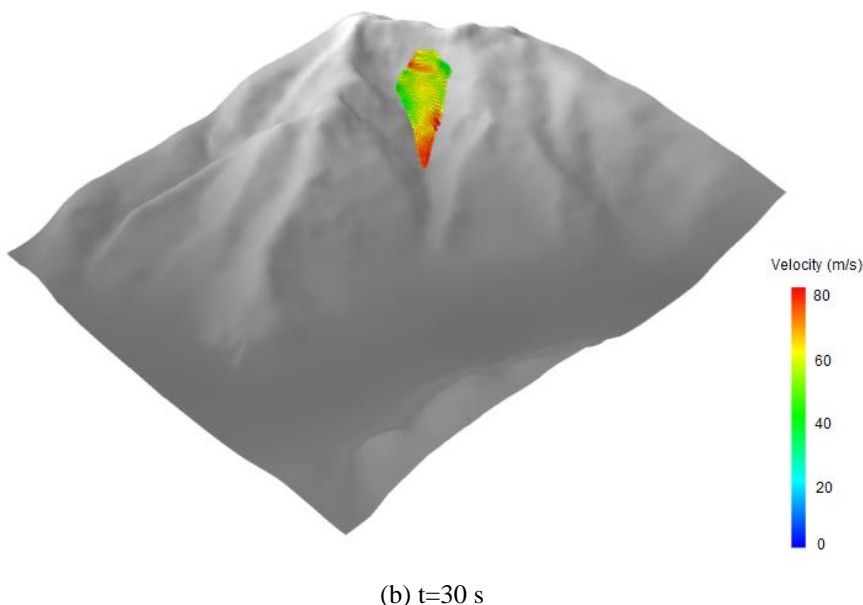

(b) t=30 s




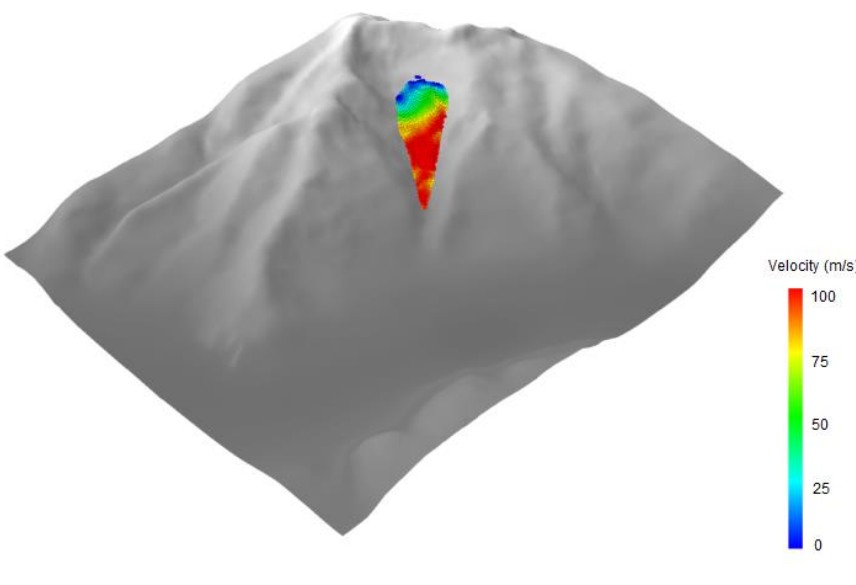

265                                        (c) t=60 s

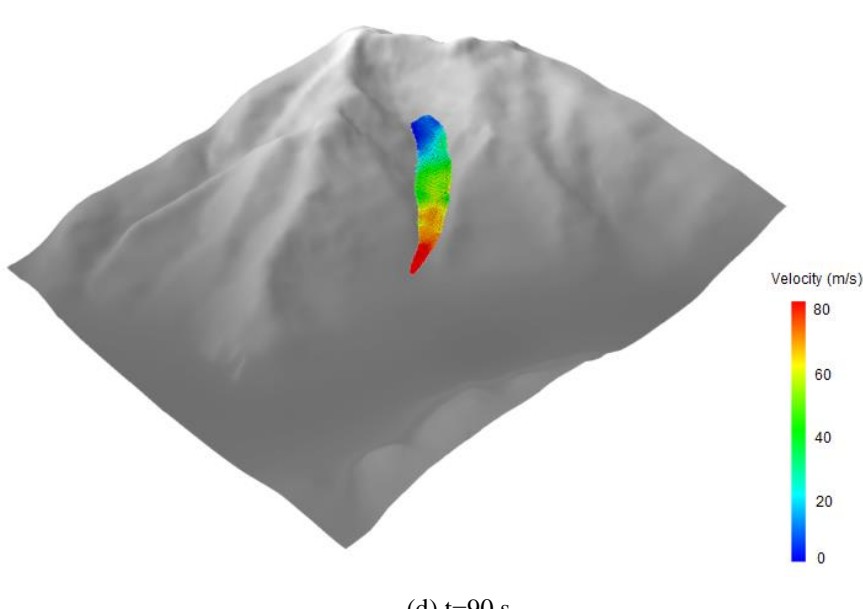

(d) t=90 s





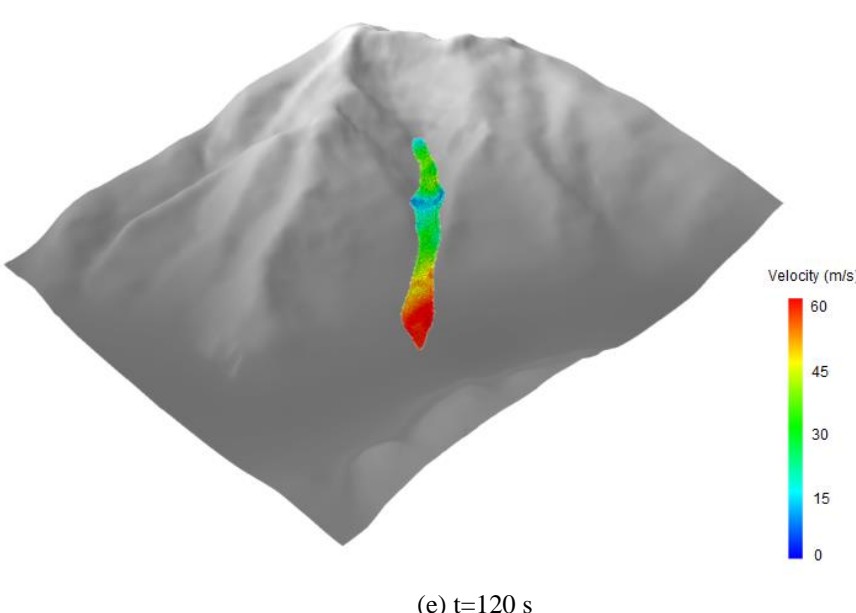

(e) t=120 s


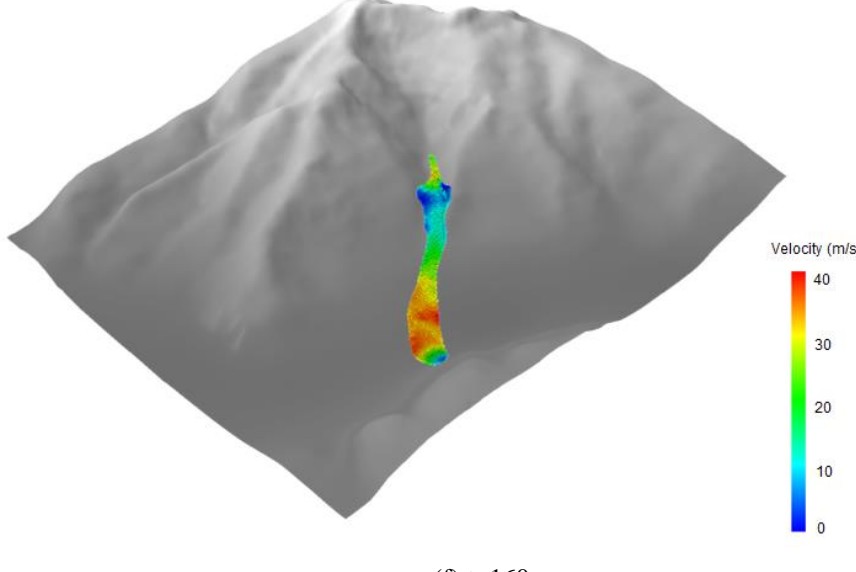

(f) t=160 s



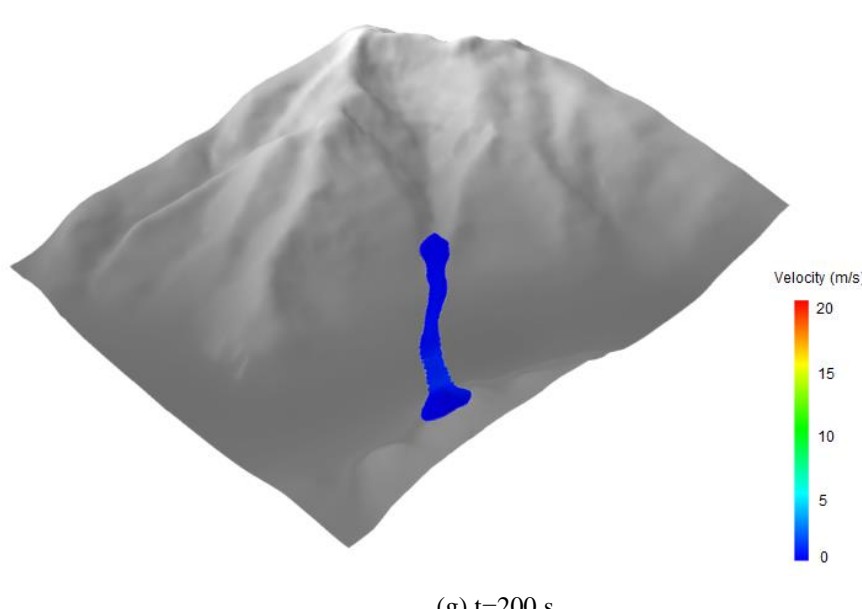

275                                    (g) t=200 s

**Figure 15: Velocity variations in the Yigong landslide.**

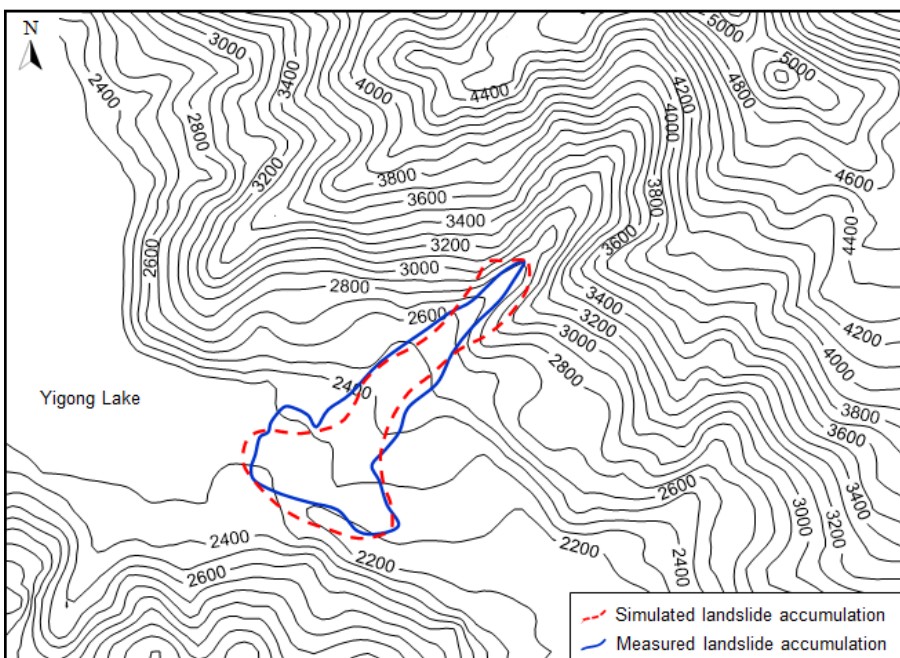

**Figure 16: Comparison of the simulated and measured deposition zone of the Yigong landslide.**



## 5 Conclusions

Rapid landslides always lead to property loss and human death all over the word. This work investigates the kinetic features of the rapid Yigong landslide on the Tibetan Plateau, China. On the basis of the SPH method, 2D and 3D numerical simulations are conducted to reproduce the motion process of the Yigong landslide. Based on the numerical results and combined with field investigation data and remote-sensing images, the kinetic characteristics of the landslide are analysed. Some conclusions can be obtained as below.

In the early stage, landslide body slides down along the deep slope in the source area. It experiences an acceleration and reaches its peak velocity (about 100 m/s). In this stage, most of the potential energy of the landslide body is converted into the kinetic energy. During the landslide propagation in the Zhamunong gully, the kinetic energy continuously dissipates due to the friction and the collision. The velocity gradually slows down in this stage. After rushing out of the Zhamunong gully, the landslide mass crashes into a mountain on the opposite bank of Yigong river and then blocks the river channel. The 290 velocity evolution of the landslide is obtained based on the numerical results, and the final landslide accumulation is predicted, which basically fits the measured data in field.

Although, the SPH model presented in this work can reproduce the motion process and analyse the kinetic characteristics of the Yigong landslide, there are still some problems need to be solved. For example, the bed material entrainment during the landslide propagation has some effect on the landslide volume and its kinetic characteristics, but it is not considered in this 295 work. On the other hand, the disintegration and fragmentation of the rock blocks is not considered in the presented SPH model, which may lead to some error during the simulation of landslide propagation. Besides, high performance parallel computing technology is quite necessary to improve the calculation efficiency in 3D modelling.

## Acknowledgments

The presented work was supported by the Program for Professor of Special Appointment (Eastern Scholar) at Shanghai 300 Institutions of Higher Learning, National Natural Science Foundation of China (grant no. 41807248, and grant no. 41530639).

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
