# Peer review of "Numerical investigation on the kinetic characteristics of the Yigong landslide in Tibet, China"

_Natural Hazards and Earth System Sciences, 2020_

## Referee Comment (RC1) · Anonymous Referee #1 · 16 Nov 2020

The writer has some concern about this paper. It seems a numerical exercise, the application of a numerical code. A question arise: which is the novelty of this numerical code when compared with other SPH numerical codes? Moreover, this code does not simulate the bed-entrainment as that of Cuomo et al. (2016). There also other deficiencies: what about the pre-event bathymetry? What about the Digital Elevation Model? Model results depends also on the data (LiDAR or photogrammetric points) by which the Digital Elevation Model is built (LiDAR, photogrammetry, see Degetto et al. 2015), the interpolation technique (Boreggio et al., 2018) and grid size (Stolz and Huggel (2008)). Information about the development of the phenomenon and post-event bathymetry are introduced without any explanations:

1) Who estimated the peak and average velocity of this rapid landslide? Which sensor

was used for measuring them? Moreover, the peak and the average values of the velocity, 100 m/s and 40 m/s respectively, seem physically not acceptable. 2) . How the post-event topography was measured?

Moreover, the reliability of a model depends on the its capability of reproducing the observed deposition pattern. The authors should compare the observed and simulated deposition depths not only the deposition area (Gregoretti et al., 2019).

Finally, some other general comments: it is strange that no erosion was observed along the flow path and that this rapid-landslide did not transform into a (muddy?) debris flow?

Other specific comments are as follows:

Lines 17-18 "This approach can provide a new way to predict hazardous areas and estimate the hazard intensity of rapid landslides." This sentence is misleading: models are used to simulate scenarios and building hazard map. Therefore, where is the novelty of this approach?

Line 216 "The simulated runout distance is about 8,500 m, which can also match the measured result very well." This sentence is useless when observed and simulate deposition pattern are compared (see figure 16) The word "accumulation" is not appropriate: use the term deposition

REFERENCES

Boreggio, M., Bernard, M. Gregoretti C. (2018) Evaluating the influence of gridding techniques for Digital Elevation Models generation on the debris flow routing modelling: A case study from Rovina di Cancia basin (North-eastern Italian Alps). Frontier in Earth Sciences, doi: 10.3389/feart.2018.00089

Degetto M, Gregoretti, C., Bernard M. (2015) Comparative analysis of the differences between using LiDAR and contour-based DEMs for hydrological modeling of runoff generating debris flows in the Dolomites. Frontiers in Earth Sciences 3:21. doi: 10.3389/feart.2015.00021.

Gregoretti, C., Stancanelli, L., Bernard, M., Degetto, M., Boreggio, M., Lanzoni, S. (2019) Relevance of erosion processes when modelling in-channel gravel debris flows for efficient hazard assessment. Journal of Hydrology, 569, 575-591; doi10.1016/j.jhydrol.2018.10.001

Stolz, A., Huggel, C. 2008. Debris flows in the Swiss National Park: the influence of different flow models and varying DEM grid size on modeling results. Landslide, 5, 311-319

---

## Referee Comment (RC2) · Anonymous Referee #2 · 24 Nov 2020

This paper analyzed the run-out process of an interesting catastrophic rockslide occurred in Tibet, China through field investigation and numerical simulation. The 2D and 3D SPH models were adopted to simulate the dynamic process of this landslide, and the Bingham model was used to describe the rheology. Then the simulated velocity and depositional characteristics were compared with field observations and measured data, and generally good results were obtained. This paper is well-written. The structure is clear, and the conclusions are reliable. The topic of this paper, which is pretty important for the mitigation of huge landslide induced disaster chain (landslide-landslide surge waves-landslide dam lake-dam break-flood chain), is definitely of interest to the readership of this Journal. Therefore, the reviewer suggests a minor revision before acceptance. The following comments are for the authors' reference.

[Figure]

ïijĹ1ïijĽ Line 14, how do you consider the effect of collision in the SPH model? Collision may contribute to the disintegration of the rock mass, and in addition, the plastic deformation caused by collision may also dissipate part of the energy. Are your models capable of depicting these effects? ïijĹ2ïijĽ There are some minor grammatical mistakes in the manuscript. Please check them carefully. For instance, in Line 27, "feathers" should be "features"? and in Line 35, "in predominantly" should be "predominantly in". In Line 250, "the results is shown" should be "the results are shown". ïijĹ3ïijĽ Line 90-95, how do you know the volume and velocities of the landslide? ïijĹ4ïijĽ Line 100, "estimated original slope surface", but how to estimate? ïijĹ5ïijĽ Line 155, are you sure the fluid is incompressible? The continuity equation is compressible, because density changes with time. And Eq. 3 is the state equation showing the relationship between density and pressure. So the fluid should be (at least) weak compressible. ïijĹ6ïijĽ Eq.3, please specify the way how you determine the parameters in this equation. And also please specify the values of these parameters in the 2D and 3D simulations. ïijĹ7ïijĽ Line 195, how do the number of particles and the kernel diameter influence the simulation results? ïijĹ8ïijĽ According to Figure 14, the maximum simulated thickness could be around 350-400m, but in Line 136, the description indicates that "the dam height is 60-120 m". Please check this inconsistency.

---

## Author Comment (AC1) · 31 Dec 2020

The authors are grateful for the reviewer' comments and suggestions. The manuscript has been revised and each point of the reviewer' comments has been incorporated and addressed. Your comments have greatly improved the quality of this manuscript and we hope the revised manuscript will be of suitable standard to be accepted for publication in your journal.

Please also note the supplement to this comment:
https://nhess.copernicus.org/preprints/nhess-2020-289/nhess-2020-289-AC1-supplement.pdf

[Figure]

**Supplement:**

**Response to the comments of reviewers for nhess-2020-289**

**Answers to Technical items for which revision is required** --- '*Numerical investigation*

*on the kinetic characteristics of the Yigong landslide in Tibet, China*'

The authors are grateful for the reviewers' comments and suggestions. The manuscript has been revised and each point of the reviewers' comments has been incorporated and addressed. Your comments have greatly improved the quality of this manuscript and we hope the revised manuscript will be of suitable standard to be accepted for publication in your journal.

**Reviewer #1:**

**1. Which is the novelty of this numerical code when compared with other SPH numerical codes? Moreover, this code does not simulate the bed-entrainment as that of Cuomo et al. (2016).**
Answer: Thank you for this comment.

a) The novelty of this numerical code is using the Open Multiprocessing (OpenMP) API to conduct the parallel implementation and improve the computational efficiency. We add some explanation in the manuscript:

"*3.2.3 OpenMP parallelism*

*To simulate the propagation of a rapid landslide across complex terrain, it is necessary to develop a three-dimensional numerical model. In the 3D SPH model, however, the computational efficiency is sharply reduced as the particle number increases. To improve the efficiency, it is necessary to parallelize the numerical code without suffering from a loss of precision.*

*The Open Multiprocessing (OpenMP) API for shared-memory programming enables loop-level parallelism by the insertion of pragmas within the source code. By adding special directives at the beginning and end of the loop, the OpenMP parallel implementation can be easily conducted. The cycles of the loop are then randomly assigned to the available threads. In the present work, the paralleled numerical code was written in FORTRAN 95 and the program was compiled using Microsoft Visual studio 2015 in a PC with the quad-core 8-thread CPU, Intel Core i7-7820HQ, and run at 2.90 GHz clock with 32 GB main memory under the Windows 10 Professional 64-bit operating system.*" (Lines 189-199)

"*To verify the performance of parallel computation, the 3D SPH modelling was carried out using different thread numbers (1, 2, 4, 6, and 8). Figure 17 shows the relationship between the average program running time and the thread number. It is obvious that the computation efficiency of the presented SPH model increases with the thread number.*" (Lines 290-292)

[Figure]

Figure 17: Relationship between average computing time and thread number in 3D SPH modelling.

b) The reason we don't simulate the bed-entrainment in this work is explained as follows:

*"Though several numerical models have been proposed to consider the entrainment effect (Cuomo et al., 2016; Li et al., 2019), it is still difficult to find an appropriate failure criterion to determine when the entrainment effect will occur. Moreover, it is also difficult to quantify the entrainment depth and volume in field investigation. Therefore, the bed entrainment effect during the propagation was not considered in the presented SPH model to simplify the simulation."* (Lines 336-340)

**2. what about the pre-event bathymetry? What about the Digital Elevation Model? Model results depends also on the data (LiDAR or photogrammetric points) by which the Digital Elevation Model is built (LiDAR, photogrammetry, see Degetto et al. 2015), the interpolation technique (Boreggio et al., 2018) and grid size (Stolz and Huggel (2008)).**

Answer: Thank you for this comment. We agree that the digital elevation model (DEM) is the fundamental input to simulate the landslide propagation, which can influence the accuracy of the model results. A simple review on the DEM generation technique is provided in the manuscript. In this study, the two-dimensional SPH model is based on the topographic profile provided in Yin (2000), and the 3D digital topographical data used in the three-dimensional SPH modelling is digitized on the contour lines (provided in Zhang, 2013) using the linear triangulation interpolation method.

*"To simulate the propagation of the flow-like landslide, the fundamental input is the topographic data, usually in the form of Digital Elevation Models (DEMs). Stolz and Huggel (2008) revealed that DEM quality and grid resolution significantly influenced the accuracy of debris flow modelling. Degetto et al. (2015) analysed the differences between using RTM-based DEM and LiDAR-based DEM for hydrological modelling of debris flows and showed that LiDAR-based DEM had relatively higher accuracy. Boreggio et al. (2018) investigated the performance of several common interpolation methods in building DEMs with the complex topography, and revealed that the interpolation algorithm had little effect on the model outcomes. In this work, the topographic profile of the Yigong landslide used in the two-dimensional SPH modelling is from Yin (2000). The 3D digital topographical data used in the three-dimensional SPH modelling is digitized on the contour lines (Zhang, 2013) using linear triangulation interpolation method."* (Lines 208-216)

*Boreggio, M., Bernard, M., Gregoretti. C.: Evaluating the Differences of Gridding Techniques for Digital Elevation Models Generation and Their Influence on the Modeling of Stony Debris Flows Routing: A Case Study From Rovina di Cancia Basin (North-Eastern Italian Alps), Frontier in Earth Sciences, 6, 89, doi: 10.3389/feart.2018.00089, 2018.*

*Degetto, M., Gregoretti, C., Bernard, M.: Comparative analysis of the differences between using LiDAR and contour-based DEMs for hydrological modeling of runoff generating debris flows in the Dolomites, Frontiers in Earth Sciences, 3: 21, doi: 10.3389/feart.2015.00021, 2015.*

*Stolz, A., Huggel, C.: Debris flows in the Swiss National Park: the influence of different flow models and varying DEM grid size on modeling results, Landslide, 5, 311-319, doi: 10.1007/s10346-008-0125-4, 2008.*

*Zhang, Y.J.: Study on dynamic characteristics of typic rock avalanche on canyon area, Ph.D. thesis, School of Naval Architecture, Ocean and Civil Engineering, Shanghai Jiao Tong University, 28 pp., 2013.*

**3. Information about the development of the phenomenon and post-event bathymetry are introduced without any explanations:**

**1) Who estimated the peak and average velocity of this rapid landslide? Which sensor was used for measuring them? Moreover, the peak and the average values of the velocity, 100 m/s and 40 m/s respectively, seem physically not acceptable.**

Answer: The runout distance of the Yigong landslide was about 8,000 m. According to eyewitness' account, the propagation time of the landslide was about 3 min (Xu et al. 2012). Therefore, it can be estimated that the average velocity was about 40 m/s. Li et al. (2020) computed the velocity process of Yigong landslide by the Massflow software, and the results showed that the peak velocity was more than 100 m/s. According to the dynamic analysis conducted by Zhang (2013), the peak and average velocity of this rapid landslide were about 111 m/s and 55 m/s, respectively. Therefore, the velocity time history predicted by the SPH model in this work is reasonable. We add some explanation in the manuscript:

"*Velocity is one of the key kinetic characteristics during the landslide propagation, which is difficult to measure in field. According to eyewitness' account, the total sliding time of the Yigong landslide was about 3 min. The runout distance was about 8,000 m. Therefore, the average sliding velocity of the landslide was estimated to be about 40 m/s. According to the dynamic analyse results (Zhang, 2013; Li et al., 2020), the maximum velocity during the landslide propagation was more than 100 m/s. Therefore, the velocity time history predicted by the SPH model in this work fits the literature data well and is reasonable and reliable.*" (Lines 327-332)

*Li, J., Chen, N.S., Zhao Y.D., Liu, M., Wang W.Y.: A catastrophic landslide triggered debris flow in China's Yigong: factors, dynamic processes, and tendency, Earth Sciences Research Journal, 24, 71-82, doi: 10.15446/esrj.v24n1.78094, 2020.*

*Xu, Q., Shang, Y., van Asch, T., Wang, S., Zhang, Z., Dong, X.: Observations from the large, rapid Yigong rock slide—debris avalanche, southeast Tibet, Canadian Geotechnical Journal, 49,589–606, doi: 10.1139/T2012-021, 2012.*

*Zhang, Y.J.: Study on dynamic characteristics of typic rock avalanche on canyon area, Ph.D. thesis, School of Naval Architecture, Ocean and Civil Engineering, Shanghai Jiao Tong University, 28 pp., 2013.*

**2) How the post-event topography was measured?**

Answer: The landslide dam broke down about two months after the landslide occurrence. Most of the landslide deposit was washed away by the flood. Therefore, we didn't measure the post-event topography by ourselves. We add some explanation in the manuscript:

"*About two months after the Yigong landslide occurrence, the landslide dam broke down, and most of the landslide deposit was washed away by the flood. Therefore, it is difficult to measure the post-event topography in field.*" (Lines 319-321)

**4. Moreover, the reliability of a model depends on the its capability of reproducing the observed deposition pattern. The authors should compare the observed and simulated deposition depths not only the deposition area (Gregoretti et al., 2019).**

Answer: We totally agree with this comment. In the two-dimensional SPH modelling, we compare the simulated deposition depths along the topographic profile with the measured results recorded in Yin (2000), as shown in Figure 14 in the manuscript. However, for the three-dimensional SPH modelling, we don't carry out the comparative analysis due to lack of measured data. We add some explanation in the manuscript as follows:

"*Figure 14 compares the simulated landslide deposition with the measured data recorded in Yin (2000). The predicted landslide deposition area is consistent with the measured data, and the simulated deposition depths along the topographic profile match the observed results well.*" (Lines 244-246)

"*Figure 16 shows the Yigong landslide deposition. The blue solid line represents the observed landslide deposition and the red dash line is the simulated results. It shows that the shape of the simulated deposition zone is basically in agreement with the observed one. The comparative analysis of deposition depths is not carried out in three-dimensional modelling due to lack of measured data, though it is important to verify the reliability of SPH model (Gregoretti et al., 2019).*" (Lines 285-289)

*Gregoretti, C., Stancanelli, L. M., Bernard, M., Boreggio, M., Degetto, M., Lanzoni, S.: Relevance of erosion processes when modelling in-channel gravel debris flows for efficient hazard assessment, Journal of Hydrology, 568, 575-591, doi: 10.1016/j.jhydrol.2018.10.001, 2019.*

**5. Finally, some other general comments: it is strange that no erosion was observed along the flow path and that this rapid-landslide did not transform into a debris flow?**

Answer: Thank you for this comment. Actually, during the propagation of the Yigong landslide, the sliding mass entrained the bed material and transformed into a debris flow. We describe the phenomenon in the manuscript as follows:

"*The high-speed sliding mass can entrain large volumes of sediments on the runout path and transform into a debris flow, which is an important feature of many rapid landslides (Gregoretti et al., 2019; Li et al., 2019). According to the field investigation conducted by Zhou et al. (2016), the bed entrainment effect during the propagation occurred at this landslide because of the high motion speed. Though several numerical models have been proposed to consider the entrainment effect (Cuomo et al., 2016; Li et al., 2019), it is still difficult to find an appropriate failure criterion to*

*determine when the entrainment effect will occur. Moreover, it is also difficult to quantify the entrainment depth and volume in field investigation. Therefore, the bed entrainment effect during the propagation was not considered in the presented SPH model to simplify the simulation.*" (Lines 333-340)

**6. Other specific comments are as follows:**

**1) Lines 17-18 "This approach can provide a new way to predict hazardous areas and estimate the hazard intensity of rapid landslides." This sentence is misleading: models are used to simulate scenarios and building hazard map. Therefore, where is the novelty of this approach?**

Answer: Thank you for this comment. This sentence is modified as "*This approach can predict hazardous areas and estimate the hazard intensity of rapid landslides.*" (Lines 17-18)

**2) Line 216 "The simulated runout distance is about 8,000 m, which can also match the measured result very well." This sentence is useless when observed and simulate deposition pattern are compared (see figure 16) The word "accumulation" is not appropriate: use the term deposition**

Answer: We agree with this comment. The sentence "The simulated runout distance is about 8,000 m, which can also match the measured result very well." is deleted. The word "accumulation" is replaced by "deposition" in the manuscript.

---

## Author Comment (AC2) · 31 Dec 2020

The authors are grateful for the reviewer' comments and suggestions. The manuscript has been revised and each point of the reviewer' comments has been incorporated and addressed. Your comments have greatly improved the quality of this manuscript and we hope the revised manuscript will be of suitable standard to be accepted for publication in your journal.

Please also note the supplement to this comment:
https://nhess.copernicus.org/preprints/nhess-2020-289/nhess-2020-289-AC2-supplement.pdf

[Figure]

[Figure]

**Supplement:**

**Response to the comments of reviewers for nhess-2020-289**

**Answers to Technical items for which revision is required** ··· '*Numerical investigation*

*on the kinetic characteristics of the Yigong landslide in Tibet, China*'

The authors are grateful for the reviewers' comments and suggestions. The manuscript has been revised and each point of the reviewers' comments has been incorporated and addressed. Your comments have greatly improved the quality of this manuscript and we hope the revised manuscript will be of suitable standard to be accepted for publication in your journal.

**Reviewer #2:**

**This paper analyzed the run-out process of an interesting catastrophic rockslide occurred in Tibet, China through field investigation and numerical simulation. The 2D and 3D SPH models were adopted to simulate the dynamic process of this landslide, and the Bingham model was used to describe the rheology. Then the simulated velocity and depositional characteristics were compared with field observations and measured data, and generally good results were obtained. This paper is well-written. The structure is clear, and the conclusions are reliable. The topic of this paper, which is pretty important for the mitigation of huge landslide induced disaster chain (landslide-landslide surge waves-landslide dam lake-dam break-flood chain), is definitely of interest to the readership of this Journal. Therefore, the reviewer suggests a minor revision before acceptance.**

**The following comments are for the authors' reference.**
**1. Line 14, how do you consider the effect of collision in the SPH model? Collision may contribute to the disintegration of the rock mass, and in addition, the plastic deformation caused by collision may also dissipate part of the energy. Are your models capable of depicting these effects?**

Answer: Thank you for this comment. In the presented SPH model, the disintegration of the rock mass caused by the collision effect cannot be considered. We add some explanation in the manuscript as follows:
"*On the other hand, high-speed sliding may lead to the disintegration of the rock mass and influence the propagation behaviour. However, this phenomenon cannot be considered in the presented SPH model.*" (Lines 340-341)

**2. There are some minor grammatical mistakes in the manuscript. Please check them carefully. For instance, in Line 27, "feathers" should be "features"? and in Line 35, "in predominantly" should be "predominantly in". In Line 250, "the results is shown" should be "the results are**

**shown".**

Answer: We are sorry for the grammatical mistakes. The manuscript has been checked carefully and the mistakes have been corrected.

**3. Line 90-95, how do you know the volume and velocities of the landslide?**

Answer: The volume of the Yigong landslide mass is about $3.0\times10^8$ m$^3$, which is cited from Shang et al. (2003). Li et al. (2020) simulated the velocity process using the Massflow software, and showed that the maximum velocity during the landslide propagation is larger than 100 m/s. According to Xu et al. (2012), the run-out time of the landslide is around 3 min, the total run-out distance is about 8,000 m. Therefore, the average sliding velocity is about 40 m/s. We add the references in the manuscript.

*"About $3.0\times10^8$ m$^3$ rock and soil slid down along the gully for about 3 min (Shang et al. 2003; Xu et al. 2012)."* (Line 92)

*"According to eyewitness' account, the total sliding time of the Yigong landslide was about 3 min. The runout distance was about 8,000 m. Therefore, the average sliding velocity of the landslide was estimated to be about 40 m/s. According to the dynamic analyse results (Zhang, 2013; Li et al., 2020), the maximum velocity during the landslide propagation was more than 100 m/s. Therefore, the velocity time history predicted by the SPH model in this work fits the literature data well and is reasonable and reliable."* (Lines 328-332)

*Li, J., Chen, N.S., Zhao Y.D., Liu, M., Wang W.Y.: A catastrophic landslide triggered debris flow in China's Yigong: factors, dynamic processes, and tendency, Earth Sciences Research Journal, 24, 71-82, doi: 10.15446/esrj.v24n1.78094, 2020.*

*Shang, Y.J., Yang, Z.F., Li, L.H., Liu, D., Liao, Q.L., Wang, Y.C.: A super-large landslide in Tibet in 2000: background, occurrence, disaster, and origin, Geomorphology 54, 225–243, doi: 10.1016/S0169-555X(02)00358-6, 2003b.*

*Xu, Q., Shang, Y., van Asch, T., Wang, S., Zhang, Z., Dong, X.: Observations from the large, rapid Yigong rock slide—debris avalanche, southeast Tibet, Canadian Geotechnical Journal, 49,589–606, doi: 10.1139/T2012-021, 2012.*

*Zhang, Y.J.: Study on dynamic characteristics of typic rock avalanche on canyon area, Ph.D. thesis, School of Naval Architecture, Ocean and Civil Engineering, Shanghai Jiao Tong University, 28 pp., 2013.*

**4. Line 100, "estimated original slope surface", but how to estimate?**

Answer: Thank you for this comment. The original slope surface is cited from Yin (2000). We add the reference in the manuscript.

*"Figure 6 shows the path profile of this landslide. In this figure, the original slope surface (blue dashed line) and the present slope surface (green solid line) are from Yin (2000)."* (Lines 100-101)

*Yin, Y.P.: Characteristics of Bomi-Yigong huge high-speed landslide in Tibet and the research on disaster prevention, Hydrogeology and Engineering Geology, 4, 8–11, 2000. (in Chinese with English abstract)*

**5. Line 155, are you sure the fluid is incompressible? The continuity equation is compressible, because density changes with time. And Eq. 3 is the state equation showing the relationship between density and pressure. So, the fluid should be (at least) weak compressible.**

Answer: We agree with the reviewer's comment. In the presented model, the landslide mass is assumed as a kind of weakly compressible fluid. We correct the expression in the manuscript:
"*In this study, the flow-like landslide is assumed as a kind of weakly compressible viscous fluid.*"
(Line 156)

**6. Eq.3, please specify the way how you determine the parameters in this equation. And also, please specify the values of these parameters in the 2D and 3D simulations.**

Answer: Thank you for this comment. In Eq.3, $\rho$ is the density calculated by the continuity equation (Eq.1). $\rho_0$ is the reference density which can be measured through laboratory tests. $c_s$ is the sound speed at the reference density, which can be set equal to ten times the maximum velocity (Zheng and Chen, 2019). $\gamma$ is a parameter which can be set to 7.0 for a good simulation of weak compressible (Zhang et al., 2020). We specify the way to determine the parameters and their values used in this paper as follows:
"*where $\rho$ is the density calculated by the continuity equation. $\rho_0$ is the reference density which can be measured through laboratory tests. $c_s$ is the sound speed at the reference density, which can be set equal to ten times the maximum velocity (Zheng and Chen, 2019). $\gamma$ is the exponent of the equation of state, and is usually set to 7.0 for a good simulation of weakly compressible (Zhang et al., 2020).*" (Lines 164-167)
"*According to Li et al. (2020), the average density of the Yigong landslide mass was about 2,000 kg/m³. The strength characteristics of the landslide mass were studied through a series of high-speed ring shear tests and rotary shear tests in the previous researches (Hu et al., 2015; Wang et al., 2017). According to the test results, the values of the c and $\varphi$ of the landslide mass can be approximately set to be 10 kPa and 20°, respectively. The sound speed cs is set to be $10v_{max}$ ($v_{max}$ is the maximum sliding velocity of the landslide mass). The parameter $\gamma$ in the equation of state is set to be 7.0 for a good simulation of weakly compressible.*" (Lines 224-230)

*Zheng, B., Chen, Z.: A multiphase smoothed particle hydrodynamics model with lower numerical diffusion, Journal of Computational Physics 382: 177–201, doi.org/10.1016/j.jcp.2019.01.012. 2019.*
*Zhang, W.J., Ji, J., Gao, Y.F.: SPH-based analysis of the post-failure flow behavior for soft and hard interbedded earth slope. Engineering Geology, 267, 105446, doi.org/10.1016/j.enggeo.2019.105446, 2020.*
*Li, J., Chen, N., Zhao, Y., Liu, M., Wang, W.: A catastrophic landslide triggered debris flow in China's Yigong: factors, dynamic processes, and tendency. Earth Sciences Research Journal, 24(1), 71–82, doi: 10.15446/esrj.v24n1.78094, 2020.*

**7. Line 195, how do the number of particles influence the simulation results?**

Answer: Thank you for this comment. According to Mao and Liu (2018), the simulation accuracy of the SPH model can be enhanced by decreasing the particle size and increasing the particle number. However, computational efficiency decreases sharply with the particle number increment (Liu and

Liu, 2003). Therefore, in the presented work, we used the appropriate number of SPH particles to achieve an appropriate balance between the computational efficiency and accuracy. We add some explanations in the manuscript as follows:

"*The number of SPH particles used in the numerical model can influence the computational efficiency and accuracy simultaneously (Liu and Liu, 2003; Mao and Liu, 2018). Therefore, to reach an appropriate balance between the computational efficiency and accuracy, 7,662 blue particles are used to represent the landslide mass and 5,906 grey image particles are used to simulate the sliding surface. The diameter of those particles is 8 m.*" (Lines 219-223)

*"Mao ZR, Liu GR. A smoothed particle hydrodynamics model for electrostatic transport of charged lunar dust on the moon surface. Computational Particle Mechanics, 2018, 5(4): 539-551.*
*Liu GR, Liu MB. Smoothed Particle Hydrodynamics: a Mesh-free Particle Method. World Scientific Press, Singapore, 2003."*